# Antibacterial and Antibiofilm Effects of Different Samples of Five Commercially Available Essential Oils

**DOI:** 10.3390/antibiotics12071191

**Published:** 2023-07-14

**Authors:** Răzvan Neagu, Violeta Popovici, Lucia Elena Ionescu, Viorel Ordeanu, Diana Mihaela Popescu, Emma Adriana Ozon, Cerasela Elena Gîrd

**Affiliations:** 1Department of Pharmacognosy, Phytochemistry, and Phytotherapy, Faculty of Pharmacy, Carol Davila University of Medicine and Pharmacy, 6 Traian Vuia Street, 020956 Bucharest, Romania; razvan.neagu@drd.umfcd.ro (R.N.); cerasela.gird@umfcd.ro (C.E.G.); 2Regenerative Medicine Laboratory, “Cantacuzino” National Military Medical Institute for Research and Development, 103 Spl. Independentei, 050096 Bucharest, Romania; popescu.diana@cantacuzino.ro; 3Department of Microbiology and Immunology, Faculty of Dental Medicine, Ovidius University of Constanta, 7 Ilarie Voronca Street, 900684 Constanta, Romania; 4Experimental Microbiology Laboratory, “Cantacuzino” National Military Medical Institute for Research and Development, 103 Spl. Independentei, 050096 Bucharest, Romania; viorel.ordeanu@prof.utm.ro; 5Department of Pharmaceutical Technology and Biopharmacy, Faculty of Pharmacy, Carol Davila University of Medicine and Pharmacy, 6 Traian Vuia Street, 020956 Bucharest, Romania; emma.budura@umfcd.ro

**Keywords:** Oregano oil, Eucalyptus oil, Rosemary oil, Clove oil, Peppermint oil, antibacterial activity, antibiofilm effect, bioactive constituents, principal component analysis

## Abstract

Essential oils (EOs) have gained economic importance due to their biological activities, and increasing amounts are demanded everywhere. However, substantial differences between the same essential oil samples from different suppliers are reported—concerning their chemical composition and bioactivities—due to numerous companies involved in EOs production and the continuous development of online sales. The present study investigates the antibacterial and antibiofilm activities of two to four samples of five commercially available essential oils (Oregano, Eucalyptus, Rosemary, Clove, and Peppermint oils) produced by autochthonous companies. The manufacturers provided all EOs’ chemical compositions determined through GC-MS. The EOs’ bioactivities were investigated in vitro against Gram-positive (*Staphylococcus aureus)* and Gram-negative bacteria (*Escherichia coli* and *Pseudomonas aeruginosa*). The antibacterial and antibiofilm effects (*ABE%* and, respectively, *ABfE%*) were evaluated spectrophotometrically at 562 and 570 nm using microplate cultivation techniques. The essential oils’ calculated parameters were compared with those of three standard broad-spectrum antibiotics: Amoxicillin/Clavulanic acid, Gentamycin, and Streptomycin. The results showed that at the first dilution (D1 = 25 mg/mL), all EOs exhibited antibacterial and antibiofilm activity against all Gram-positive and Gram-negative bacteria tested, and MIC value > 25 mg/mL. Generally, both effects progressively decreased from D1 to D3. Only EOs with a considerable content of highly active metabolites revealed insignificant differences. E. coli showed the lowest susceptibility to all commercially available essential oils—15 EO samples had undetected antibacterial and antibiofilm effects at D2 and D3. Peppermint and Clove oils recorded the most significant differences regarding chemical composition and antibacterial/antibiofilm activities. All registered differences could be due to different places for harvesting the raw plant material, various technological processes through which these essential oils were obtained, the preservation conditions, and complex interactions between constituents.

## 1. Introduction

Essential oils (EOs) are highly concentrated plant derivatives defined based on their physicochemical properties [1]. The Eos’ chemical composition includes phenolic compounds, terpenes, terpenoids, phenylpropanoids, and other aliphatic and aromatic constituents. European Pharmacopoeia (Ph. Eur.) defines EOs as odorous products, usually of complex composition, obtained from a botanically defined plant raw material by steam distillation, dry distillation, or a suitable mechanical process without heating [2]. Generally, EOs have high contents (20–70%) of two or three major phytoconstituents; other compounds are quantified in trace concentrations [3]. The pharmacological potential of EOs has been extensively explored: ranging from antioxidant [4,5], anti-inflammatory [6,7], immunomodulatory [8,9], antinociceptive [10,11], antiulcer [12,13], anticancer [14,15], insecticidal [16,17], larvicidal [18,19], anthelmintic [20,21], antiviral [22,23], antibacterial and antibiofilm [24,25,26,27,28] properties. 

Currently, substantial growth in the EOs’ general use worldwide has been reported in various domains: the food industry, fragrances, aromatherapy, and cosmetics, personal care, spa and relaxation, home care, and healthcare (pharmaceuticals and nutraceuticals) [29]. Especially, aromatherapy (implying—professionals—aromatherapists—individual customers) has gained extensive applications in numerous countries. 

Therefore, the global market for essential oils is anticipated to have a continuous expansion, with regional differences. For example, in the USA, its size is expected to approximately double in 2030 compared to 2022 [30]. The USA Food and Drug Administration (FDA) divides EOs into three categories:Cosmetics—Products intended to clean the body (except for soap);Household items/Other—Fragrance products, like scented candles, household cleaners, and air fresheners;Drugs—Products intended for therapeutic use that can treat or prevent various diseases or affect the body structure or function [31,32,33].

The EOs from doTERRA (Pleasant Grove, UT, USA) [34] are commercialized with the label CPTG (Certified Pure Therapeutic Grade). However, the FDA did not regulate essential oils as foods or dietary supplements. Any EO product cannot be marketed with the following mention: “It is intended to treat, prevent, cure or mitigate any disease or other health condition”—even when scientific research supports the claims’ validity.

In Europe, the EOs industry growth is promoted by the European Federation of Essential Oils (EFEO) [35]. Currently, EFEO is discussing with the European Commission and the EU Parliament to amend or introduce legislation concerning essential oils. In contrast, due to numerous bioactivities, the European Medicinal Agency (EMA) considers EOs as herbal preparations and as active pharmaceutical ingredients (API) in two groups of herbal products [36]: Herbal medicinal products (HMPs), both for human and veterinary use;Traditional herbal medicinal products (THMPs) for human use.

Thus, EMA established rigorous quality documentation for all manufacturers, and competent national authorities can refer to one unique set of information concerning registered EOs as HMPs/THMPs when evaluating marketing applications [37,38]. In 2022, EMA revised the “Guideline on specifications: test procedures and acceptance criteria for herbal substances, herbal preparations, and herbal medicinal products/traditional herbal medicinal products” [39]. When essential oils are used as APIs of HMPs, the quality guidelines require an analytical characterization of the raw material. Moreover, according to the monograph “Herbal Drugs” from Ph. Eur., other tests must be performed on the essential oils [36]. According to GMP standards, the manufacturing process is another point, implying the quality of water used for the EOs distillation from fresh plants. The composition of essential oils should be within the Ph. Eur. Monograph’s limits. Compliance with the ISO standards and the Ph. Eur. limits is critical for revealing adulterated EOs, evaluating correct plant material, chemotypes, and provenance, and recognizing changes during fabrication and storage [40].

All manufacturers must have suitable quality documentation for EOs, following EMA regulations. However, it is difficult to achieve all documents [36] when farmers or very small companies are implied in the manufacturing processes.e Therefore, substantial differences could be recorded between the same essential oil samples from different suppliers due to a lack of regulation, numerous companies involved in EOs production, and continuous development of online sales. Thus, measuring the metal content of 34 EOs from various manufacturers, Iordache et al. [41] identified Peppermint oil with Hg levels over six times higher than Ph. Eur. permissible limits. Vargas Jentzsch et al. [42] investigated commercial Clove essential oil samples. They found three adulterated samples containing benzyl alcohol and vegetable oil [42]. Recently, Pierson et al. [43] tested 31 EO samples purchased online by evaluating their compliance with ISO standards; they found that more than 45% did not pass the test, and more than 19% were diluted with various solvents (propylene and dipropylene glycol, triethyl citrate, or vegetable oil) [43]. In a previous study, Brun et al. [44] investigated 10 commercially available Tea Tree essential oils, finding that only 5 samples had significant antimicrobial activity [44].

The present study aims to explore the antibacterial and antibiofilm effects of five commercially available EOs—well-known for their phytotherapeutic applications—against Gram-positive and Gram-negative bacteria. Four essential oils (Eucalyptus oil, Rosemary oil, Clove oil, and Peppermint oil) are registered by EMA as HMPs for human and veterinary use [45], having a periodically updated monography (Table 1) [46]. Moreover, they have individual monographs in Ph. Eur., indicating the bioactive constituents’ concentration limits, which are the basis of EMA regulation. 

Only Oregano oil is authorized as a feed additive for animal species [47], and these data follow ISO 13171:2016 from International Organization for Standardization, Geneva, Switzerland [48]. Phytogenic feed additives (phytobiotics) are currently used in traditional European animal healthcare [49]. The effect of Oregano oil dietary supplementation in poultry on production parameters, intestinal villi height, and broiler breast’s antioxidant capacity is well studied [50,51]. 

The antibacterial effects of essential oils investigated in our study are implied in all their therapeutical benefits, as mentioned in Table 1.

**Table 1 antibiotics-12-01191-t001:** The main applications of essential oils, according to the European Medicinal Agency.

	Essential Oil Name	Botanical Name	Therapeutic Area/ Applications
1.	*Origani aetheroleum* (Oregano oil)	*Origanum vulgare* ssp. hirtum (Link) Ietsw.	Feed additive for specific animal species [47]
2.	*Eucalypti* *aetheroleum* (Eucalyptus oil)	*Eucalyptus globulus* Labill. *Eucalyptus polybractea* R.T. Baker. *Eucalyptus smithii* R.T. Baker.	Pain and inflammation Cough and cold [52]
3.	*Rosmarini* *aetheroleum* (Rosemary oil)	*Rosmarinus officinalis* L.	Circulatory disorders Gastrointestinal disorders [53]
4.	*Caryophylli floris* *aetheroleum* (Clove oil)	*Syzygium aromaticum* (L.) Merr. et L.M. Perry, syn. *Eugenia caryophyllus* (Spreng.) Bullock et S.G. Harrison	Mouth and throat disorders [54]
5.	*Menthae piperitae aetheroleum* (Peppermint oil)	*Mentha piperita* L.	Pain and inflammation Skin disorders and minor wounds Cough and cold Gastrointestinal disorders [55]

Our study novelty consists of a different design. We selected five essential oils with well-known antibacterial effects [40,41,42,43,44,45,46,47] to facilitate the detection of the potential differences between the results obtained from the tested samples and those mentioned in the literature data. As ordinary consumers, we checked various Romanian markets (including online suppliers) and selected only four that concomitantly provided the essential oils’ chemical composition. We compared the bioactive metabolite contents with those mentioned in each monograph in Ph. Eur. 10 and ISO 13171:2016 according to EMA’s current regulation. Then, the antibacterial and antibiofilm effects of two to four samples of each EO were evaluated, correlating the data obtained with the bioactive compounds. The results were compared with the most known data from the scientific literature. In addition, a complex statistical analysis was performed to support our results. 

## 2. Results

### 2.1. Antibacterial and Antibiofilm Activity on S. aureus

The percentage values of antibacterial and antibiofilm efficacy of essential oils against Gram-positive bacteria (*S. aureus*) tested, compared to standard antibiotics, are displayed in Table 2. 

Table 2 shows that, at 50 µg/mL, AMC had remarkable antibacterial efficacy (*ABE =* 96.83%), while both aminoglycosides exhibited a good one (*ABE* > 85%). All EOs recorded good antibacterial activity (*ABE* > 75%) against *S. aureus* at D1 = 25 mg/mL. OEOs and CEO1 proved to have considerable antibacterial potentials (*ABE* = 92.76, 90.40, 91.26%) like the AMC one and higher than GEN and STR. The CEO3, PEO3, and PEO4 had the lowest *ABE* (78.80, 79.73, 79.83%). *S. aureus* sensitivity commonly decreases directly proportional to EOs concentration. Only a few EOs recorded good antibacterial effects at all D1, D2, and D3 dilutions: EEOs and REO1 (*ABE%* = 75–89%). The anti-staphylococcal effect slowly diminished at progressive dilutions (as in the case of REO2) or intensely decreased (OEOs, CEOs, PEOs 1 and 2). PEOs 3 and 4 exhibited antibacterial effects only at 25 mg/mL. As an overview, we could appreciate that the MIC value for all EOs tested is higher than 25 mg/mL. In addition, no significant differences (*p* > 0.05) between the antibacterial efficacy values of the tested samples at D1 were recorded in the case of OEOs, EEOs, and REOs (Table 2). At D1, the *ABE* values of all three CEOs were statistically significant (91.26, 84.80, 78.80%, *p* < 0.05), and of PEOs (PEO1 and PEO2 vs. PEO3 and PEO4: 87.50 and 85.23% vs. 79.73 and 79.83%, *p* < 0.05).

Data from Table 2 also show that all standard antibiotics and EOs have antibiofilm activity at D1 and D2. 

At D1, AMC shows a moderate *ABfE,* and both aminoglycosides report a satisfactory one. For EOs, it decreases from “very good” (OEOs, CEO1, and PEO1) to good (CEOs 2 and 3 and PEO2), moderate (EEO1), and satisfactory (EEO2, REOs, PEOs 3 and 4). Excepting OEOs, the samples of each EO recorded significant differences (*p* < 0.05, Table 2). The highest differences were recorded in the case of PEOs 1 and 2 vs. PEOs 3 and 4 (91.13 and 89.23% vs. 32.17 and 38.73%, Table 2). Moderate differences in *ABfE* values at D1 were also registered in CEOs (95.40, 77.13, 81.77%, Table 2).

At D2, most EOs recorded significant differences in antibiofilm effect compared to D1 (*p* < 0.05), except CEO1 and CEO3 (95.87 vs. 95.40%, respectively, 83.50 vs. 81.77%, *p* > 0.05, Table 2). 

At D3 (0.25 mg/mL), the antibiofilm activity registered the lowest values for all standard antibiotics and a significant part of EOs samples (*ABfE* < 50%). It was not detected for EEOs and CEOs 2 and 3. The CEO1 and PEOs 1 and 2 showed *AbfE* values > 50% (56.87, 77.20, and 81.87%, respectively, Table 2).

### 2.2. Antibacterial and Antibiofilm Activity on E. coli

The percentage values of antibacterial and antibiofilm efficacy of essential oils against Gram-negative bacteria *E. coli*, compared to standard antibiotics, are displayed in Table 3. Table 3 shows that at D1 = 50 µg/mL, AMC exhibits a very good *ABE*, while both aminoglycosides display a good one (*ABE* < 90%). 

All EOs inhibited *E. coli* strains growing at D1 = 25 mg/mL; the antibacterial effect decreased from "very good" (OEO1 and CEO1) to moderate (PEO2–4). Most EOs recorded good inhibitory effects against *E. coli* (Table 3). Statistically significant differences were recorded in CEOs—between CEO1 and CEOs 2 and 3 (92.47% vs. 78.10 and 77.80%, *p* < 0.05)—and PEOs—PEO1 vs. PEOs 2, 3, and 4 (79.00% vs. 72.50, 72.77, and 71.30%, *p* < 0.05, Table 3).

At D2, most EOs samples registered significant statistical differences compared to D1; only EEOs and REOs showed similar *ABE* values (Table 3). The PEOs 2, 3, and 4 (72.50 vs. 43.87%, 72.77 vs. ND, and 71.30 vs. 4.60%) and CEOs 1, 2, and 3 (92.47 vs. 46.27%, 78.10 vs. 34.70%, and 76.80 vs. 18.50%) displayed the highest differences (Table 3).

At D3, the antibacterial activity of CEOs 2 and 3 and PEOs 3 and 4 was undetected. At the same time, in the case of other EOs, it was substantially decreased compared to D2: OEO1 (13.40 vs. 79.40%), PEO1 (45.57 vs. 70.00%), PEO2 (29.80 vs. 43.87%), and CEO1 (22.30 vs. 46.27%). Only *ABE* of EEO1 showed insignificant differences (83.73 vs. 84.23%, *p* > 0.05, Table 3). From standard antibiotics, AMC reported an intense diminution of *ABE* value (19.60 vs. 84.47%, Table 3).

The antibiofilm activity of standard antibiotics and EOs differs significantly at D1 (Table 3). AMC and GEN show a high *ABfE*, while STR has a moderate one. However, OEOs and CEO1 significantly inhibited *E. coli* biofilm formation (95.13, 95.63, 95.10%, *p* > 0.05), while CEOs 2 and 3 had a good antibiofilm effect (82.20, 78.50%, *p* > 0.05). Table 2 also shows that EEOs have a moderate antibiofilm action (61.73 and 52.33%, *p* < 0.05), followed by REOs and PEO1 with satisfactory *ABfE* values (49.23 and 30.07%, *p* < 0.05, and 34.23%). The PEOs 2, 3, and 4 recorded the lowest inhibitory activity on *E. coli* biofilm formation (2.90, 3.10, and 4.50%, respectively, Table 3). 

Insignificant differences between D1 and D2 were observed at CEO1, STR, and AMC (Table 3). In the case of other EOs and GEN, the *ABfE* values significantly decreased (*p* < 0.05, Table 3). On CEOs 2 and 3 and all PEOs, the antibiofilm activity was undetected at D2 and D3 (Table 3). At D3, the other five EOs (OEO2, EEOs, and REOs) reported undetected antibiofilm activity on *E. coli*; in addition, the *ABfE* values of OEO1 and CEO1 substantially decreased (2.93 and 11.20%, respectively, Table 3).

### 2.3. Antibacterial and Antibiofilm Activity on P. aeruginosa

The percentage values of antibacterial and antibiofilm efficacy of essential oils against *P. aeruginosa*, compared to standard antibiotics, are displayed in Table 4.

Table 4 shows that all standard antibiotics (GEN, STR, and AMC) exhibited very good inhibitory activity on *P. aeruginosa* strains growing at 50 µg/mL (91.80, 92.80, and 91.40%, *p* > 0.05), AMC having a lower one than both aminoglycosides. At 25 mg/mL, both OEOs and CEO1 (92.00, 91.40, and 93.60%, *p* > 0.05) revealed an antibacterial efficacy similar to standard antibiotics, while CEOs 2 and 3 reported a moderate one (67.80 and 59.60%, *p* < 0.05, Table 4). All the other EOs displayed good antibacterial efficacy against *P. aeruginosa.* Only between CEO samples (CEO1, 2, and 3) were registered significant differences at D1, *p* < 0.05 (Table 4). 

At D2, EEOs and REOs did not show significant differences compared to D1: *ABE* = 84.70–87.90%, *p* > 0.05 (Table 4). The same observation is available for PEO2 (79.20 vs. 86.10%, *p* > 0.05, Table 4). Most EOs (OEOs, CEOs, and PEOs 1 and 4) reported a significant diminution in *ABE* values (*p* < 0.05, Table 4). The PEO 2 antibacterial activity at D2 was undetected (Table 4). 

At D3, EEOs maintained the same antibacterial activity without significant differences reported to D1 and D2 *(ABE* = 84.93–87.90%, *p* > 0.05, Table 4). The REOs 1 and 2 (75.73 and 66.10% vs. 84.70 and 85.77%, *p* < 0.05) and PEOs 1 and 2 (69.70 and 61.40% vs. 77.44 and 79.20%, *p* < 0.05) displayed more significant differences than D2 but not as high (Table 4). The OEO2, CEO1, and standard antibiotics exhibited low antibacterial activity at D3, undetected at CEOs 2 and 3, and PEOs 3 and 4 (Table 4).

Table 4 reveals that OEOs, EEOs, REOs, and CEO1 displayed substantial antibiofilm activity at D1 and D2 (*ABE* > 90%, *p* > 0.05). Their biofilm formation inhibition is higher than standard antibiotics (Table 4). At D1, *ABfE* values for CEO2 and CEO3 are <50% of CEO1 (43.90 and 44.30% vs. 94.80%, *p* < 0.05). It strongly diminished at D2 (7.10 and 16.40% vs. 92.40%, *p* < 0.05, Table 4). Moreover, PEO4 has *ABfE* value higher than PEOs 1, 2, and 3: 85.80% vs. 74.60 and 73.20% (*p* < 0.05), and 79.90% (*p* > 0.05); at D2, the antibiofilm activity of PEOs 3 and 4 was undetected, and that of PEOs 1 and 2 was similar (68.80 vs. 65.90%, *p* > 0.05, Table 4). 

At D3, REOs and EEOs maintained a good antibiofilm effect *(ABfE* > 85%), while the other EOs reported a high diminution: OEOs (27.77 and 8.50%, *p* < 0.05), CEOs 1 and 2 (16.00 vs. 4.10%, *p* < 0.05), and PEOs 1 and 2 (57.20 vs. 48.40%, *p* < 0.05). Moreover, CEO3 and PEOs 3 and 4, like standard aminoglycosides, did not show antibiofilm activity at D3 against *P. aeruginosa* (Table 4). 

### 2.4. Data Analysis

Pearson correlation applied on OEOs showed a substantial positive correlation between carvacrol and linalool and antibacterial effect against all three bacteria and antibiofilm activity on *S. aureus* and *P. aeruginosa* (*r* = 0.999, *p* < 0.05). Thymol, *p*-cymene, γ-terpinene, and α-terpinene strongly correlate with antibiofilm activity in *E. coli (r* = 0.998, *p* < 0.05).

Principal Component Analysis (PCA) was used to evaluate the correlation between the bioactive compounds and antibacterial and antibiofilm efficacy of EOs on Gram-positive and Gram-negative bacteria.

The PCA-Biplot from Figure 1 shows the correlation between the previously mentioned variable parameters for two EOs (EEO and REO) with common constituents (eucalyptol, α-pinene, β-pinene, camphene, limonene, *p*-cymene, and β-myrcene). The Correlation Matrix from Appendix A shows evidence of substantial correlations between several secondary metabolites and antibacterial and antibiofilm effects of EEOs and REOs. Eucalyptol, limonene, p-cymene, and γ-terpinene substantially correlate with antibacterial effects against *E. coli* (*r* = 0.931, *r* = 0.937, *r* = 0.944, *r* = 0.913, *p* > 0.05). At the same time, they show a good and moderate correlation with antibiofilm activity on *E. coli* (*r* = 0.882, *r* = 0.803, *r* = 0.759, *r* = 0.731, *p* > 0.05). They also moderately correlate with antibiofilm effects on *S. aureus (r* = 0.633, *r* = 0.711, *r* = 0.658, *r* = 0.746, *p* > 0.05) and *P. aeruginosa (r =* 0.728, *r* = 0.662, *r* = 0.503, *r* = 0.612, *p* > 0.05). Moreover, limonene, p-cymene, and γ-terpinene moderately correlate with antibacterial activity against *S. aureus* (*r* = 0.610, *r* = 0.759, *r* = 0.701, *p* > 0.05). α-Phellandrene displays a considerable correlation with antibacterial activity against *E. coli* (*r* = 0.873, *p* > 0.05) and a good and moderate one with antibiofilm effects on *P. aeruginosa, E. coli*, and *S. aureus* (*r* = 0.811, *r* = 0.806, *r* = 0.734, *p* > 0.05). Figure 1 also shows the place of each EEO and REO sample, correlated to chemical composition and antibacterial and antibiofilm activities.

Figure 2 displays the correlations between bioactive constituents and CEOs’ antibacterial and antibiofilm effects. 

Thus, eugenol substantially correlates with all antibacterial and antibiofilm activities: *E.c. ABE* and *P.a. ABfE* (*r* = 0.999, *p* < 0.05), *E.c. ABfE* (*r* = 0.986, *p* > 0.05), *P.a. ABE* and *S.a. ABfE* (*r* = 0.982, *p* > 0.05), *S.a. ABE* (*r* = 0.897, *p* > 0.05). It has the highest content compared to other volatile compounds quantified in CEOs and shows the most substantial correlations with antibacterial and antibiofilm activities. 

Eugenyl acetate and β-caryophyllene correlate moderately with *S.a. ABE* (*r* = 0.611, *r* = 0.760, *p* > 0.05). β-caryophyllene displays a low correlation (r = 0.543, r = 0.559, *p* > 0.05) with *E.c. ABfE* and *P.a. ABE*. Finally, the PCA-Biplot shows each CEO sample’s place considering these variable parameters; it provides evidence that CEO1 is substantially active regarding antibacterial and antibiofilm effects. All detailed data are found in Appendix A.

Figure 3 shows the correlations between bioactive constituents and PEOs’ antibacterial and antibiofilm effects. From bioactive constituents, eucalyptol shows a remarkable and significant statistical correlation with antibacterial activity against *P. aeruginosa (r =* 0.995, *p* < 0.05). It is also considerably correlated with *ABE* against *E. coli* (*r* = 0.830, *p* > 0.05) and moderately associated with *ABE* against *S. aureus* (*r* = 0.716, *p* > 0.05) and *ABfE* in *S. aureus* and *E. coli* (*r* = 0.593, *r* = 0.685, *p* > 0.05). However, eucalyptol negatively correlates with *ABfE* in *P. aeruginosa* (*r* = −0.803, *p* > 0.05). Concomitantly, menthol is considerably correlated with both *ABE* and *ABfE* in *S. aureus* (*r* = 0.847, *r* = 0.826, *p* > 0.05) and *E. coli* (*r* = 0.680, *r* = 0.754, *p* > 0.05). Moderate correlations were evidenced between menthone and pulegone and antibiofilm activity against *P. aeruginosa* (*r* = 0.736, *r* = 0.503, *p* > 0.05); both previously mentioned constituents and menthofuran are negatively correlated with *P.a. ABE* (*r* = −0.579, *r* = −0.818, *r* = −0.617, *p* > 0.05). Isomenthone and neomenthol with *ABfE* on *S. aureus* (*r* = 0.708, *r* = 0.550, *p* > 0.05). Moreover, isomenthone is moderately correlated with antibacterial activity against *S. aureus* (*r* = 0.636, *p* > 0.05). 

Finally, considering all discussed variable parameters—extensively described in Appendix A—Figure 3 shows the place of each PEO sample, thus explaining all differences between them and supporting the results.

The correlation matrix from Appendix A and Figure 4A highlights a strong correlation between antibacterial and antibiofilm effects against both Gram-negative bacteria, *P. aeruginosa* (*r* = 0.912, *p* > 0.05) and *E. coli (r* = 0.760, *p* > 0.05). On *S. aureus,* both effects are poorly correlated (*r* = 0.425, *p* > 0.05) when we evaluate this effect on the entire EOs group. However, for all EOs and standard antibiotics, the antibacterial activity against *S. aureus* is considerably associated with that against *E. coli (r* = 0.895, *p* < 0.05) and moderately with that against *P. aeruginosa* (*r* = 0.628, *p* < 0.05). Antibacterial effects against Gram-negative bacteria also show a moderate correlation (*r* = 0.878, *p* < 0.05). All data are statistically significant (*p* < 0.05). Generally, antibiofilm activities on all bacteria tested are poorly correlated. As an overview, on the first dilution (D1 = 25 mg/mL), only OEOs show similar percentage values of antibacterial and antibiofilm effects.

The registered data from Results are summarized in Figure 4A, evidencing the place of essential oils reported to both *ABE* and *ABfE* against all bacteria tested. 

In a simplified manner, the dendrogram obtained by Agglomerative Hierarchical Clustering (AHC) from Figure 4B and Appendix A shows how EO samples act similarly. Figure 4B shows that PEO1 acts similarly to PEO2, PEO3 to PEO4, and CEO2 to CEO3. On the other hand, both OEOs have similar effects at D1. At the same time, OEO2 acts closely to CEO1. The same observation is available on EEOs 1 and 2 and REOs 1 and 2.

### 2.5. Correlations between EOs Chemical Constituents and Their Antibacterial and Antibiofilm Effects

Knowing each EO’s chemical composition, the correlations between bioactive constituents and antibacterial and antibiofilm effects were analyzed to explain the differences between the corresponding samples.

Figure 5A indicates that only *p*-cymene’s contents in both OEO samples are included in Standard limits (4–10%). OEO1 displays 69.60% carvacrol, 5.60% γ-terpinene, and 2.60% thymol; all values are in the ISO 13171:2016 limits. In contrast, OEO2 has significantly lower carvacrol content (44.40% < ISO St.Min.) and considerably higher thymol (3×) and γ-terpinene (2×) concentrations than ISO St.Max. Furthermore, linalool was detected in OEO1 and α-terpinene in OEO2, unmentioned in ISO 13171:2016. Figure 5B shows that OEOs’ antibacterial and antibiofilm activities against Gram-positive and Gram-negative bacteria are similar. All constituents contribute to these effects. OEO1 has the highest carvacrol content, but OEO2 contains thymol and γ-terpinene more than ISO 13171:2016 maximal limits. Their effects are augmented by *p*-cymene, linalool, and α-terpinene; the last two volatile compounds are missing in the standard. 

Figure 6A shows that the eucalyptol content in EEO2 is lower than EEO1 and Ph. Eur.Min., while the α-pinene content is higher. Both EEOs have three compounds unmentioned in Ph. Eur. 10 (*p*-cymene, γ-terpinene, and α-phellandrene). Figure 6B reveals that both REOs have eucalyptol content substantially higher than Ph. Eur.Max. (48.10 and 42.05 > 25%); β-pinene ones are slightly increased than Ph. Eur. 10 maximal value (6.84 and 6.89 > 6.00%). α-Pinene (12.29 and 13.35%), camphor (9.23 and 10.26%), and camphene (4.42 and 5.28%) are significantly lower than Ph. Eur.Min. (18, 13, and 8%). EEOs and REOs display considerable antibacterial effects on all bacteria tested (Figure 6C,D). The antibiofilm activities are substantial on *P. aeruginosa* and moderate on *S. aureus* and *E. coli*. Eucalyptol is the primary metabolite responsible for it in both EOs; its content in REOs was higher than maximal limits from Ph. Eur. 10. Its activity is potentiated by other compounds (limonene, *p*-cymene, and γ-terpinene) in both EOs and α-phellandrene in EEOs.

Figure 7A shows that all three CEO samples contain eugenol and eugenyl acetate in the Ph. Eur. 10 range (75–88%, respectively, 4–13%). CEO1 and CEO3 have the highest eugenol content (80.00–80.06%). The eugenyl acetate content decreases in order: CEO2, CEO1, and CEO3 (13.27 > 10.25 > 5%). CEO3 did not have β-aryophyllene. Some aspects concerning PEOs’ chemical composition were observed, analyzing Figure 7B. Thus, all PEOs contain menthone and menthyl acetate in the Ph. Eur 10 limits (14–32% and, respectively, 2.80–10%). PEO1 has the highest contents of menthol, isomenthone, and eucalyptol (39.27, 4.75, and 7.70%), while PEO4 shows the greatest ones for menthone (28.45%), menthofuran (3.89%), and pulegone (2.90%), and PEO2 for menthyl acetate (6.01%) and neomenthol (4.42%). PEO3 contains only four bioactive compounds: menthol (<30%), menthone, menthyl acetate, and eucalyptol; PEO4 has no eucalyptol. Moreover, two constituents mentioned in Ph. Eur. 10 (carvone and isopulegol) were not found in PEOs samples. At the same time, neomenthol, contained by PEO1, PEO2, and PEO4, did not appear in *Menthae piperitae aetheroleum* monograph from Ph. Eur. 10.; the same observation is available for menthofuran (3.89%) and pulegone (2.90%) and PEO2 for menthyl acetate (6.01%) and neomenthol (4.42%). PEO3 contains only four bioactive compounds: menthol (< 30%), menthone, menthyl acetate, and eucalyptol; PEO4 has no eucalyptol. Moreover, two constituents mentioned in Ph. Eur. 10 (carvone and isopulegol) were not found in PEOs samples. At the same time, neomenthol, contained by PEO1, PEO2, and PEO4, did not appear in *Menthae piperitae aetheroleum* monograph from Ph. Eur. 10. 

The antibacterial activity of CEO and PEO samples is different. CEO1, due to substantial eugenol content, acts like OEOs (Figure 4). The antibacterial and antibiofilm effects decrease in order of CEO1, CEO2, and CEO3. In addition, eugenyl acetate and β-caryophyllene synergistically act with eugenol. CEO2 and CEO3 display similar antibacterial and antibiofilm potential, significantly lower than CEO1.

Figure 4 and Figure 7D show that PEO1 and PEO2 and, respectively, PEO3 and PEO4, act similarly. The primary metabolites implied in antibacterial and antibiofilm effects are menthol, menthone, menthyl acetate, and eucalyptol. They synergistically act with the others in lower content. 

All previously mentioned observations are available for D1 = 25 mg/mL. At the following two dilutions (D2 = 2.5 mg/mL and D3 = 0.25 mg/mL), the antibacterial and antibiofilm effects could remain similar, slowly decrease, substantially diminish, or be undetected. 

On *S. aureus* and *P. aeruginosa,* the antibacterial activity of both EEOs did not report significant differences (Table 2 and Table 4), while on *E. coli,* only EEO1 reported similar *ABE* values (Table 3). Regarding antibiofilm activity, EEO1 and both REOs recorded similar effects (*p* > 0.05) at all three dilutions against *P. aeruginosa.*

On *S. aureus*, the following EOs exhibited no significant differences in antibiofilm activity between D1 and D2: PEOs 1 and 2, CEOs 1 and 3, and OEO1; on *E. coli,* EEO2, and REOs, regarding antibacterial activity; finally, on *P. aeruginosa,* REOs and PEOs *(ABE)* and PEO1, CEO1, EEO2, OEOs *(ABfE).* In contrast, at D2 and D3, the antibacterial activity of PEOs 3 and 4 against *S. aureus* was undetected. Similar observations are available on *E. coli:* PEO3 has no antibacterial effects, and CEO2, CEO3, and all PEOs did not record antibiofilm activities (Table 3). On *P. aeruginosa,* PEOs 3 and 4 did not show antibiofilm activity, and only PEO3 had no antibacterial efficacy (Table 4).

## 3. Discussion

According to European Pharmacopoeia, an EO is an odorous product with complex composition obtained by steam distillation, dry distillation, or other mechanical processes without heating from a botanically defined plant raw material. The Eos’ separation from the aqueous phase uses a physical process without significantly affecting their composition. This definition reveals that other extraction procedures involving different solvents lead to extracts, not EOs. Due to increased demands and insufficient regulation, the adulteration of essential oils became a common practice along supply chains, generating safety concerns in the EOs industry [56]. The EOs adulteration could be performed using various methods: a cheaper oil addition [57] to the original one (e.g., corn mint to Peppermint oil), EOs’ dilution with vegetable oils, and synthetic phytochemicals’ inclusion [51] in the original EO [58,59,60]. Thus, supplementary quality control measures should be taken to ensure safety for human use [61,62]. Because aromatherapy is the most crucial application of essential oils, the best way to test the properties of various samples of the same EO is by using biological systems [63]. Pierson et al. [38] recently signaled the potential consumer vulnerability to neroli, mandarin, and bergamot essential oils purchased online. 

Many patients request pharmacists’ counseling, aiming to know how to select from numerous types of commercially available essential oils the most suitable ones for therapeutic purposes. In Romania, the most known manufacturers mention the following data on each EO packaging and leaflet [64,65].

The scientific name of the raw plant.100% Pure—It is not mixed with other essential oils, synthetic components, plant oils, or mineral oils and has no chemical solvents.100% Natural—It is obtained by steam distillation.100% Verified—It is biochemically and botanically defined.Notification number at the National Service for Medicinal, Aromatic Plants and Hive Products [66], National Research and Development Institute for Food Bioresources—IBA Bucharest [67] according to the National Regulation on Food Supplements.

On some manufacturers’ websites, the GC-MS analysis reports are available, indicating the chemical composition of the EOs [68,69]. The others provide GC-MS reports upon request. Therefore, putting ourselves in the patient’s place, we have chosen well-known, most studied, and widely used EOs in therapy due to their antibacterial properties precisely so that we can relate our results to the data from the literature. We purchased and analyzed several samples of the same EO from different suppliers. All investigated EOs samples were manufactured by four autochthonous companies and regularly commercialized in pharmacies, pharma markets [70], and online markets. We requested the GS-MS reports with chemical composition to compare them with the one from the Ph. Eur. 10 monograph, or ISO standard, as stipulated by the EMA regulation. The manufacturers provided each EO’s chemical composition, and all statistical analyses were performed with available data [71]. Next, we analyzed their influence on antibacterial and antibiofilm effects against Gram-positive and Gram-negative bacteria tested using the microdilution method. Three decimal dilutions, 25, 2.5, and 0.25 mg/mL, were used for EOs; the values were established based on the data used in previously published studies from the scientific literature. We selected the other three (D1 = 50 µg/mL, D2 = 5 µg/mL, and D3 = 0.5 µg/mL) for standard antibiotics, considering their MIC values on bacterial strains according to CLSI and EUCAST [71].

Regarding the OEOs’ chemical composition, it was observed that OEO1 contains all four constituents (carvacrol, *p*-cymene, γ-terpinene, and thymol) in the suitable Ph. Eur. limits (Figure 5A). OEO2 has a lower carvacrol content, but other metabolites in augmented concentrations than OEO1: thymol content—seven times higher, γ-terpinene—three-point-five times higher, and *p*-cymene—two times higher. A similar thymol concentration was quantified by Salehi et al. in OEO from Greece [72]. Moreover, another two compounds, unmentioned in Ph. Eur., were found in the GC-MS report: linalool and β-caryophyllene, in up to 2% concentration. They could contribute to the antibacterial effects due to complex interaction with the other bioactive metabolites [44]. Our results confirm these aspects (Figure 5B).

The main constituents of OEO are carvacrol and thymol, which have solid pharmacological potential, including antibacterial, anti-inflammatory, and antioxidant activities. At the same time, carvacrol and thymol, obtained by chemical synthesis, could be adulterants of Oregano oil [48]. Gavaric et al. [73] reported the additive antibacterial effect of thymol and carvacrol against tested bacterial strains (*S. aureus*, *E. coli*, *Salmonella infantis,* and *Bacillus cereus*). *p*-Cymene is the biological precursor of carvacrol; when used alone, it exhibits a lower antibacterial effect than carvacrol; however, it synergistically acts with carvacrol against *E. coli* [74] and *B. cereus* [75]. Furthermore, carvacrol can form a chimera with DNA [76], while thymol reduces enterotoxins A, B, and α-hemolysin secreted by *S. aureus* isolates [77]. 

Thus, OEO could be considered a broad-spectrum natural antibiotic [78,79]. Investigating the antibacterial mechanism against MRSA, Cui et al. proved that OEO affects bacterial wall permeability, leading to an irreversible depletion [76]. It can inhibit bacterial respiratory metabolism (perturbing the tricarboxylic acids cycle) and the expression of MRSA’s crucial pathogenic factor PVL.

In the present study, both OEOs display similar antibacterial and antibiofilm activities, reporting the highest effects compared to all EOs investigated. Their inhibitory activity against all bacteria tested is similar to Amoxicillin-Clavulanic acid. Our results were similar to those obtained in other studies, which confirm both activities (antibacterial and antibiofilm) on *S. aureus* [80], *E. coli* [81], and *P. aeruginosa* [82,83].

Our PCA analysis, separately performed on OEOs, reported a strong statistically significant correlation between the bioactive constituents and *ABE,* respectively, *ABfE,* thus justifying OEOs’ records. Carvacrol and linalool were substantially correlated (*r* = 0.999, *p* < 0.05) with *ABE* and *ABfE* against *S. aureus* and *P. aeruginosa* and only with *ABE* against *E. coli.* In contrast, thymol, linalool, α-, and γ-terpinene evidenced the same correlation with antibiofilm activity in *E. coli.* Numerous studies from scientific literature reported the antibacterial and antibiofilm properties of all previously mentioned phenolic compounds [84,85,86,87,88,89,90,91,92,93,94,95].

For all bacteria tested, MIC > 25 mg/mL. Other studies indicated various MIC values: 0.16–0.32 mg/mL against MRSA [96], >3.2 mg/mL against *E. coli* [97,98], and 0.16–0.64 mg/mL against *P. aeruginosa* [96]. 

Of six metabolites mentioned in the EEO monograph from Ph. Eur. 10, only four (eucalyptol, α-pinene, α-phellandrene, and limonene) appear in the available GC-MS reports (Figure 6A). Their concentration in EEO1 is included in regulatory limits. EEO2 has a lower content of eucalyptol (61.45% vs. 79.73%) and a six times higher concentration of α-pinene (18.77% vs. 3.60%). The samples have small contents of other compounds: myrcene (in EEO1), *p*-cymene, and γ-terpinene (in EEO1 and EEO2).

Bachir et al. [99,100], reported the antibacterial efficacy of *Eucalypti aetheroleum* against *S. aureus* due to its phytoconstituents (eucalyptol, linalool, β-pinene). Moreover, its bactericidal effect against *E. coli* and *P. aeruginosa* was reported [101]. The EEO’s bioactive constituent responsible for the antibiofilm activity is eucalyptol [102,103,104]. Therefore, Eucalyptus oil penetrates the biofilm matrix, interfering with the essential constituents’ synthesis and the metabolic processes of the biofilm. EEO has synergistic antibacterial activity against Gram-positive bacteria, while against Gram-negative ones, it is additive [105]. Furthermore, eucalyptol obtained by chemical synthesis could be mixed with Eucalyptus oil for adulteration [106].

The present study proved *Eucalypti aetheroleum*’s appreciable antibacterial effectiveness against *S. aureus*, *E. coli*, and *P. aeruginosa* (MIC > 25 mg/mL). The EEO’s MIC against MRSA varies between 0.032–307 mg/mL [28]. Mulyaningsih et al. [107] evidenced antibacterial activity against *E. coli* with a MIC > 4 mg/mL. In comparison, Van et al. reported a median MIC of 27.26 mg/mL against *P. aeruginosa* isolates [108]. However, EEOs recorded a moderate antibiofilm efficacy on *S. aureus* and *E. coli* strains and a substantial one against *P. aeruginosa,* like OEOs. Minimal differences were registered between both tested samples, with EEO1 acting higher than EEO2. Both common constituents are considerably correlated with antibacterial/antibiofilm activities. Thus, data analysis performed exclusively on EEOs indicated a strong statistically significant correlation between eucalyptol, limonene, and *p*-cymene and both antibacterial and antibiofilm activities against *S. aureus* and *E. coli*; at the same time, α-pinene is substantially correlated with *ABE* and *ABfE* against *P. aeruginosa* (*r* = 0.999, *p* < 0.05). The antibacterial and antibiofilm activities of these phenolic metabolites were revealed by other previously published studies [109,110,111,112,113,114,115,116,117,118,119,120].

Compared to Ph. Eur. data, REOs have around two times higher eucalyptol content (48.10% in REO1 and 42.50% in REO2) and no verbenone (Figure 6B). In addition, REO2 contains 4.45% β-caryophyllene; α and β-Pinene camphor, borneol, and camphene were substantially correlated with *ABE* and *ABfE* on *P. aeruginosa* (*r* = 0.999, *p* < 0.05). Eucalyptol has a similar correlation with *ABE* against *E coli*; camphor and camphene are correlated with *ABE* and *ABfE* against *S. aureus;* pinene beta only correlates with *ABfE* on *S. aureus*. These compounds are known for their antibacterial and antibiofilm effects [121,122,123,124,125,126,127,128,129].

Previous studies [97,130,131,132,133] evidenced the antibacterial effects of *Rosmarini aetheroelum* against *S. aureus* and *E. coli*. Santoyo et al. [133] highlighted the antibacterial efficacy of REO against *P. aeruginosa* due to the bioactive constituents, camphor, borneol, and verbenone. Rosemary oil also inhibits *P. aeruginosa* biofilm formation [134].

Our results show that *Rosmarini aetheroleum* had significant antibacterial and antibiofilm efficacy against *P. aeruginosa*. The antibacterial activity is similar for all Gram-positive and Gram-negative bacteria tested (*ABE* > 80.00%, MIC > 25 mg/mL). Other studies reported MIC values of 6.2–25 mg/mL against *S. aureus*, 12.5–25 mg/mL against *E. coli*, and 50 mg/mL against *P. aeruginosa* [135]. REO1 was slightly more active than REO2, but no significant differences were recorded between the two samples’ effects. 

Moreover, the synthetic equivalents of their main components, eucalyptol and camphor, could be used for Rosemary oil adulteration [136].

Generally, their chemical composition corresponds to Ph. Eur.; CEO1 has the highest eugenol content, followed by CEO3 and CEO2 (Figure 7A). However, CEO3 has the lowest eugenyl acetate concentration and no β-caryophyllene; all constituents are highly and moderately correlated with antibacterial and antibiofilm activities [137,138,139,140,141,142,143]. 

Xu et al. [144] highlighted the antibacterial efficacy of *Caryophylli aetheroleum* against *S. aureus* (with a MIC value = 0.625 mg/mL). They hypothesized that the volatile oil destroys the cell wall and membranes, causing loss of vital intracellular materials, resulting in bacterial death. The volatile oil also penetrates the cytoplasmic membrane and inhibits the normal synthesis of DNA and proteins necessary for bacterial growth. Yadav et al. [145] reported the antibiofilm effect of Clove oil on *S. aureus* attributed to eugenol. It inhibits biofilm formation, interrupts intercellular connections, detaches pre-existing biofilms, and kills bacteria in biofilms. Synthetic eugenol is also used for Clove oil adulteration [42]. 

The MIC value in the present study for all bacteria tested is >25 mg/mL. Generally, the MIC values of CEO against *S. aureus* vary in the range of 0.52–1.04 mg/mL [146]. 

Burt et al. [147] evidenced the antibacterial efficacy of *Caryophylli aetheroleum* against *E. coli*; the CEO’s MIC value belonged to the range of 0.64–1.28 mg/mL [146]. Another study by Kim et al. [148] reported the antibiofilm efficacy of Clove oil against *E. coli* due to eugenol inhibitory activity on biofilm formation. 

The CEO’s antibacterial efficacy against *P. aeruginosa* is also demonstrated [149], with a MIC of 4.9 mg/mL [150]. Moreover, the antibiofilm activity of Clove oil is due to its main bioactive compounds, eugenol and eugenyl acetate [151]. 

The present study reports a few differences between the three CEO samples. 

Thus, CEO1 showed substantial antibacterial and antibiofilm efficacy against all Gram-positive and Gram-negative bacteria tested (with *ABE* and *ABfE* values > 91.80%). 

CEO2 and CEO3 proved good antibacterial and antibiofilm effectiveness against *S. aureus* and *E. coli*. However, significant differences were registered in their effects against *P. aeruginosa,* exhibiting moderate and satisfactory antibacterial and antibiofilm activity. 

All PEOs contain menthone and menthyl acetate in the Ph. Eur. limits (Figure 7B). Only PEOs 1, 2, and 4 have suitable menthol content; PEO3 is under 30%. Isomenthone, neomenthone, menthofuran, and pulegone were not detected in PEO3, while PEO4 does not have eucalyptol. These phytochemical aspects can explain the differences between PEOs 1–4 bioactivities. PEO1 and PEO2 have the highest concentrations of menthol, menthyl acetate, isomenthone, and eucalyptol; these constituents considerably correlate with antibacterial and antibiofilm activities. Moreover, synthetic menthol could substitute Peppermint in adulterant oil [152].

Generally, in Peppermint oil, the association of menthol, menthone, limonene, neomenthol, carvone, and eucalyptol with other minor constituents appears to induce a synergistic antibacterial activity. A recent study [153] evaluated the antibacterial activity of volatile oil obtained from *Mentha piperita* L. leaves on MDR strains from hospitalized patients. The authors used bacterial cell lines (ATCC) and isolates of *S. aureus, E coli,* and *P. aeruginosa,* proving PEOs’ bactericidal effects against all microorganisms.

Li et al. [154] evidenced that *Menthae aetheroleum* (with a high content of carvone, menthone, isomenthone, neomenthol, menthol, and menthyl acetate) has a significant antibacterial effect against *S. aureus* [155]. All tested samples of Peppermint oil showed appreciable anti-staphylococcal efficacy. Kang et al. [156] showed that PEO inhibits the biofilm of *S. aureus* by altering the permeability and integrity of bacterial cell membranes. Peppermint oil significantly inhibits biofilm formation and inactivates the mature biofilm [157]. 

Alamoti et al. [158] proved the antibacterial efficacy of *Menthae aetheroleum* against *E. coli* due to pulegone content. 

Peppermint oil also inhibits *P. aeruginosa* [159], showing substantial antibiofilm activity [160,161]. 

All four PEO samples investigated in the present study had remarkable antibacterial effects against Gram-positive and Gram-negative bacteria, with no significant differences (MIC > 25 mg/mL). They recorded the highest *ABE* (>85.00%) on *P. aeruginosa* and *S. aureus* (*ABE >* 79.70%). On *E. coli,* the PEOs’ antibacterial efficacy was good to moderate, in the range of 71.30–79.00%; PEO1 shows the highest effect. Evaluating the antibacterial effect of EO from *Mentha piperita* L. against MDR bacterial strains, Muntean et al. reported the following MIC values range: 5–20 mg/mL on *S. aureus,* 10–20 mg/mL on *E. coli*, and 20–40 mg/mL on *P. aeruginosa* [153].

Regarding the antibiofilm activity, the PEOs displayed considerable effects on *P. aeruginosa, ABfE =* (73.20–85.80%). On *E. coli*, PEOs registered the lowest effects: *AbfE* = (2.90–34.20%). The most significant differences were highlighted in the antibiofilm efficacy evaluation against *S. aureus.* The obtained data show that PEO1 and PEO2 have a substantial antibiofilm activity *AbfE* = (89.20–91.00%). Concomitantly, PEO3 and PEO4 exhibited a poor antibiofilm effect (32.10–38.70%). 

Some significant observations are available to corroborate all obtained results and compare them with the literature data. As an overview, the samples with different manufacturers of the same essential oil showed similar activities; only Clove and Peppermint oils showed higher differences. Compared to other studies’ results, the low differences between EEO and REO samples appear not to influence the antibacterial effects. Oregano oils showed a substantially lower antibacterial activity against *S. aureus* and *P. aeruginosa;* the MIC values (>25 mg/mL) are significantly higher than those mentioned in literature data: 0.16–0.32 mg/mL against MRSA and 0.16–0.64 mg/mL against *P. aeruginosa.* The same observation is available for Clove oils against *S. aureus* and *E. coli;* the MIC values from literature data are substantially lower (0.52–1.04 mg/mL, respectively, 0.64–1.28 mg/mL) than those obtained in the present study. Moreover, appreciable differences were recorded in the chemical composition of CEO samples. Notable differences in bioactive constituents’ content were registered in PEOs samples, resulting in high antibacterial and antibiofilm effects variations. However, the literature data indicates large ranges of MIC variation against all bacteria tested.

All these differences could be explained by the significant variation of EOs’ chemical composition, bacterial strains selected, and technical aspects implied in microbiological assays used.

## 4. Materials and Methods

### 4.1. Materials

All chemicals and reagents were of analytical grade. Poly (ethylene glycol)-block-poly (propylene glycol)-block-poly (ethylene glycol) (Poloxamer 407) and Crystal Violet (Gentian Violet) were purchased from Sigma-Aldrich Chemie GmbH (Schnelldorf, Germany). 

Gentamicin^®^ 80 mg/2 ml (GEN) injectable solution was supplied by KRKA (Novo mesto, Slovenia). Antibiotice SA (Iași, Romania) provided Streptomycin (Strevital^®^) 1 g (STR) powder for an injectable solution and Amoxicillin-Clavulanic acid (Amoxiplus^®^) 1.2 g (AMC) [162] powder for an injectable solution.

Gram-positive (*S. aureus*) and Gram-negative (*E. coli* and *P. aeruginosa*) bacteria were obtained from the sub-collection of the Experimental Microbiology Laboratory of the “Cantacuzino” National Military Medical Institute for Research and Development, Bucharest. Other recently published studies used these strains for antibacterial activity screenings [163,164]. Sanimed International Impex SRL (Calugareni, Romania) was the Muller–Hinton culture media supplier.

The laboratory equipment consisted of an EnSight Multimode Plate Reader (PerkinElmer, Waltham, Massachusetts, USA), an adjustable incubator (Memmert GmbH + Co.KG, Büchenbach, Germany), a microplate shaking incubator (Heidolph Instruments GmbH & Co. KG, Schwabach, Germany), microbiological hood class II with laminar flow (Jouan SA, Pays de la Loire, France), Evoqua double-water distiller (Evoqua Water Technologies GmbH, Barsbüttel, Germania), and an electronic scale (Ohaus Corporation, Parsippany, NJ, USA). The NUNC™ MaxiSorp™ 96-well plates were supplied from Electron Microscopy Sciences (Hatfield, PA, USA).

Five commercially available essential oils were purchased from Romanian markets, 2–4 samples for each EO, noted with *1, 2, 3*, and *4*:○*Origani aetheroleum 1, 2* (Oregano essential oil, OEO);○*Eucalypti aetheroleum 1, 2* (Eucalyptus essential oil, EEO);○*Rosmarini aetheroleum 1, 2* (Rosemary essential oil, REO);○*Caryophylli aetheroleum 1, 2, 3* (Clove essential oil, CEO);○*Menthae aetheroleum 1, 2, 3, 4* (Peppermint essential oil, PEO).

Four different companies produced these EOs: Fares S.A. (Orastie, Romania), Aromateria (Targu Mures, Romania), DVR Pharm (Brasov, Romania), and Ecoterapia (Bucharest, Romania). They provided their chemical composition, determined through GC-MS (Figure 1 and Figure 2).

### 4.2. Antibacterial Activity

The current method was adapted from [165,166]. It involved the cultivation of bacteria in 96-well microplates with Muller–Hinton medium with EOs samples and incubation at 37 °C for 24 h.

#### 4.2.1. Inoculum Preparation

The direct colony suspension method (CLSI) was used for preparing the bacterial inoculum. First, bacterial colonies selected from a 24 h agar plate were suspended in an MHA medium. The bacterial inoculum was accorded to the 0.5 McFarland standard, measured at Densimat Densitometer (Biomerieux, Marcy-l’Étoile, France) with around 10^8^ CFU/mL (CFU = colony-forming unit). 

#### 4.2.2. Sample Preparation

The samples were O/W emulsions prepared with an essential oil concentration of 30% *w*/*w*; the emulsifier was Poloxamer 407 5% in water, as previously mentioned [167].

Each emulsion was diluted with double distilled water to achieve the final concentration of each EO stock solution (25 mg/mL). 

#### 4.2.3. Standard Antibiotic Solutions Preparation

All antibiotic drug solutions were prepared with double distilled water, the final stock solution concentration being 0.5 mg/mL. 

#### 4.2.4. Microdilution Method

All successive steps were performed in a laminar flow; in 96-well plates, we performed serial dilutions, adapting the protocol described by Gómez-Sequeda et al. [166] and detailed in our recently published study [163]. All well plates were incubated for 24 h at 37 °C.

After incubation, the antibacterial efficacy of essential oils was determined by reading the absorbance values using the EnSight Multimode Plate Reader and calculated according to Sandulovici et al. [163]. 

### 4.3. Antibiofilm Activity

The method was adapted from [168,169] and detailed in our recently published article [163]. After incubation, the bacterial biofilm production was evidenced by staining with 0.1% Gentian Violet after removing the culture medium, washing twice with sterile distilled water, and drying at room temperature under airflow. After dye removal, the microplates were dried at 50 °C for 60 min. The dye incorporated in bacterial cells that formed the biofilm was solubilized with 95% ethanol for 10 min under continuous stirring at 450 rpm. 

### 4.4. Quantification and Interpreting of Antibacterial and Antibiofilm Activities

The antibacterial and antibiofilm effect of essential oils was determined by reading the absorbances separately on each experimental variant using the EnSight Multimode Plate Reader. Depending on the measurement needs, the operator selected and modified the wavelengths (562 nm for antibacterial effect and 570 nm for antibiofilm activity). The final absorbance value is the arithmetic mean of the instrument software’s 100 readings per second/well-made automatically [122]. 

The calculation formula is presented in the following equation (Equation (1)) and detailed in the Appendix A.
(1)$$Efficacy\;\%=100-\frac{mean\;of\;Sample\;absorbance\;value\;}{Reference\;absorbance\;value}\times 100$$

The obtained results were compared to standard antibiotics.

Interpretation of antibacterial (*ABE%—bacterial growth inhibition%*) and antibiofilm (*ABfE%—biofilm formation inhibition%*) efficacy was quantified on conventional arithmetic intervals: very good efficacy: ≥90%, good efficacy: 75–89%, moderate efficacy: 50–74%, satisfactory: 25–49% and unsatisfactory: 0–24%. 

### 4.5. Data Analysis

The analyses were performed in triplicate; the results are expressed as a mean ± Standard Deviation (SD). 

The statistically significant differences were determined using Anova single factor from Microsoft 365 Excel^®^ v.2023 (Microsoft Corporation, Redmond, WA, USA). The statistically significant values were marked in Table 1, Table 2 and Table 3 with superscripts [122].

The correlations between variable parameters [169] were examined through principal component analysis [170] performed with XLSTAT 2023.1.4. by Lumivero (Denver, CO, USA) using Pearson correlation. 

The statistical significance was established at *p* < 0.05 [170].

## 5. Conclusions

All essential oils exhibited antibacterial and antibiofilm activities on the first decimal dilution against all Gram-positive and Gram-negative bacteria tested, and MIC value > 25 mg/mL. The obtained results could be explained by the significant variation of EOs chemical composition, bacterial strains selected, and technical aspects implied in microbiological assays used.

Generally, both effects significantly decreased proportionally with serial dilutions when the concentration of the bioactive compounds recorded a progressive diminution. Only EOs with a considerable content of highly active metabolites revealed insignificant differences. *E. coli* showed the lowest susceptibility to all commercially available essential oils—15 EO samples had undetected antibacterial and antibiofilm effects at the following two dilutions. Only EOs with a considerable content of highly active metabolites revealed insignificant differences at all decimal dilutions. The essential oils with many bioactive compounds in moderate contents recorded a substantial diminution of antibacterial potential. 

Samples with different provenance of the same essential oil showed similar activities; thus, both OEOs and CEO1, EEOs and REOs, CEO2 and CEO3, PEO1 and PEO2, and PEO3 and PEO4 acted similarly. Clove and Peppermint oils showed higher variations due to the bioactive compounds’ different contents. The most substantial differences in bioactive constituents’ contents were registered in PEO samples, leading to high antibacterial and antibiofilm effects variations. 

All these differences could be due to different places for harvesting the raw plant material, various technological processes through which these essential oils were obtained, the preservation conditions, and complex interactions between constituents. 

Further research will quantify the bioactive constituents of each EO sample, extending, at the same time, the therapeutical properties investigation.

## Figures and Tables

**Figure 1 antibiotics-12-01191-f001:**
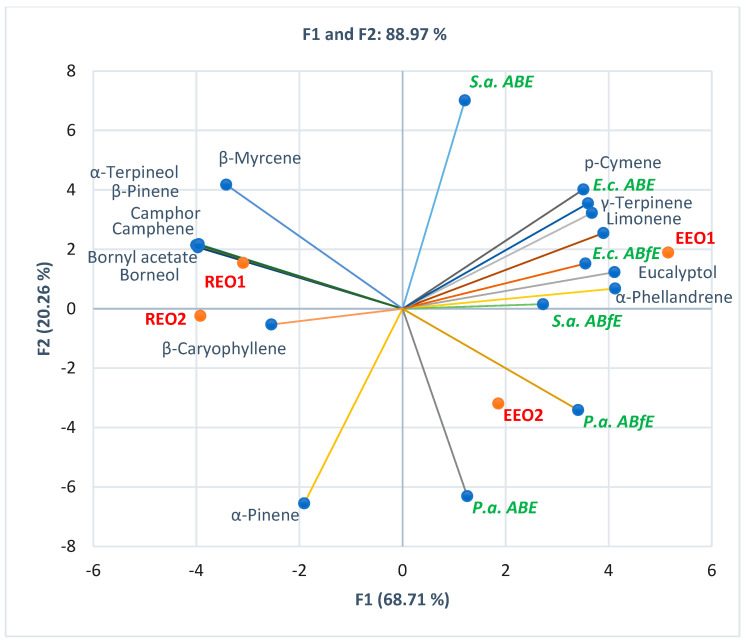
PCA-Biplot displays the correlations between bioactive constituents and antibacterial and antibiofilm effects on Gram-positive and Gram-negative bacteria in each EEO and REO sample. EEOs 1–2—Eucalyptus essential oils from two different manufacturers; REOs 1–2—Rosemary essential oils from two different manufacturers; *S.a.—S. aureus, E.c.—E. coli, P.a.—P. aeruginosa, ABE*—Antibacterial efficacy, *ABfE*—Antibiofilm efficacy.

**Figure 2 antibiotics-12-01191-f002:**
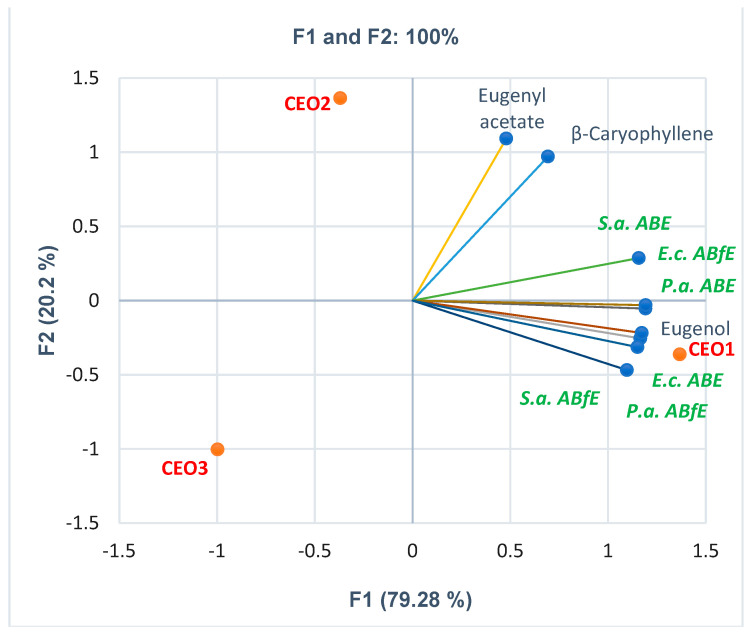
PCA-Biplot displays the correlations between bioactive constituents’ content and antibacterial and antibiofilm effects on Gram-positive and Gram-negative bacteria in each CEO sample. CEOs 1–3—Clove essential oils from three different manufacturers; *S.a.—S. aureus, E.c.—E. coli, P.a.—P. aeruginosa, ABE*—Antibacterial efficacy, *ABfE*—Antibiofilm efficacy.

**Figure 3 antibiotics-12-01191-f003:**
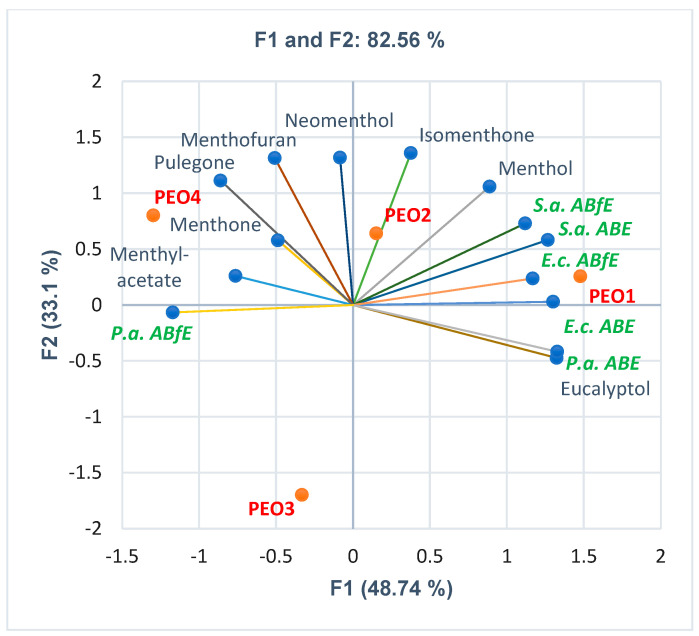
PCA-Biplot displaying the correlations between bioactive constituents’ content and antibacterial and antibiofilm effects on Gram-positive and Gram-negative bacteria in each PEO sample; PEO1–4—Peppermint essential oil from four different manufacturers; *S.a.—S. aureus, E.c.—E. coli, P.a.—P. aeruginosa, ABE*—Antibacterial efficacy, *ABfE*—Antibiofilm efficacy.

**Figure 4 antibiotics-12-01191-f004:**
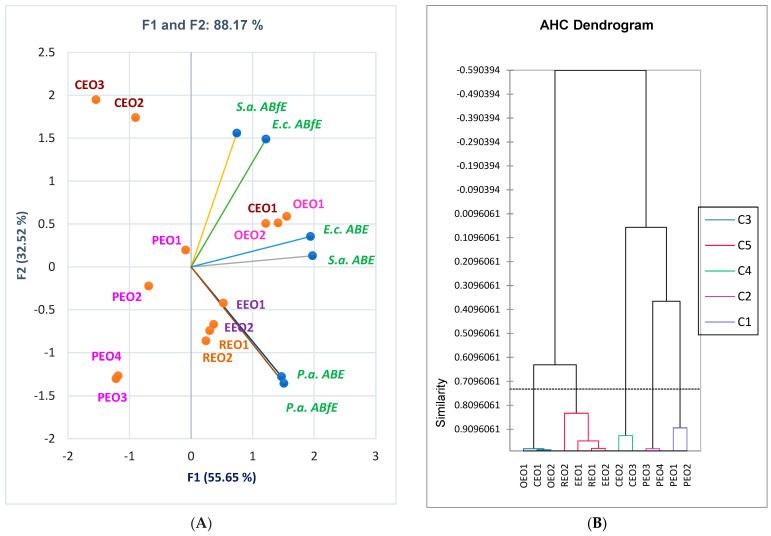
(**A**) PCA-Biplot displays the antibacterial and antibiofilm efficacy of essential oils and antibacterial drugs against Gram-positive and Gram-negative bacteria. (**B**) AHC-Dendrogram. C1–C5 —clusters: C1 (PEO1, PEO2), C2 (PEO3, PEO4), C3 (CEO1, OEO1, OEO3), C4 (CEO2, CEO3), C5 (REO1, REO2, EEO1, EEO2). GEN—Gentamicin; STR—Streptomycin; AMC—Amoxicillin-Clavulanic acid; OEOs 1–2—Oregano essential oils from two different manufacturers; EEOs 1–2—Eucalyptus essential oils from two different manufacturers; REOs 1–2—Rosemary essential oils from two different manufacturers; CEOs 1–3—Clove essential oils from three different manufacturers; PEOs 1–4—Peppermint essential oils from four different manufacturers; *S.a.—S. aureus, E.c.—E. coli, P.a.—P. aeruginosa, ABE*—Antibacterial efficacy, *ABfE*—Antibiofilm efficacy.

**Figure 5 antibiotics-12-01191-f005:**
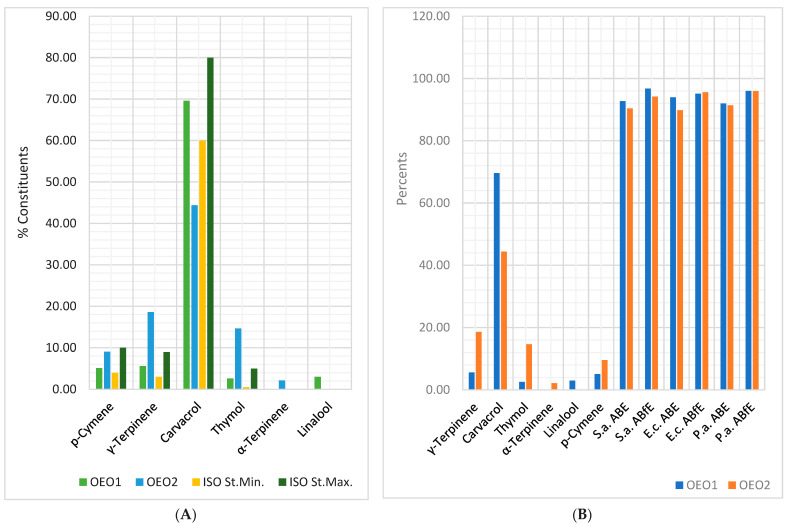
(**A**) Chemical composition of OEO samples provided by the manufacturers, compared to ISO 13171:2016 standard. (**B**) Bioactive constituents and antibacterial and antibiofilm effects of OEOs. The correlation between chemical constituents and antibacterial and antibiofilm activities of *Oregano oil* samples. S.a.—*S. aureus*, E.c.—*E. coli*, P.a.—*P. aeruginosa*, *ABE*—Antibacterial efficacy, *ABfE*—Antibiofilm efficacy, ISO St.Min./Max.—the constituents’ content limits from ISO 13171:2016 standard.

**Figure 6 antibiotics-12-01191-f006:**
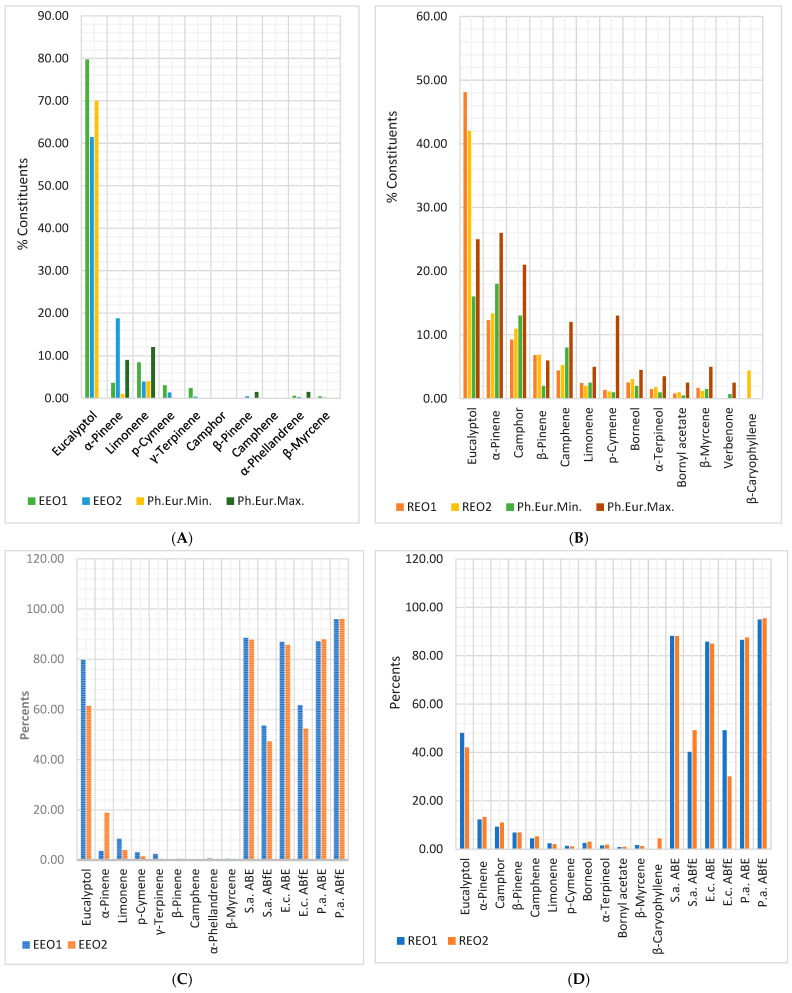
(**A**,**B**) Chemical composition of EEOs and REOs samples provided by the manufacturers, compared to Ph. Eur. 10 limits (Min/Max). (**C**,**D**) Bioactive constituents and antibacterial and antibiofilm effects of EEOs (**C**) and REOs (**D**).

**Figure 7 antibiotics-12-01191-f007:**
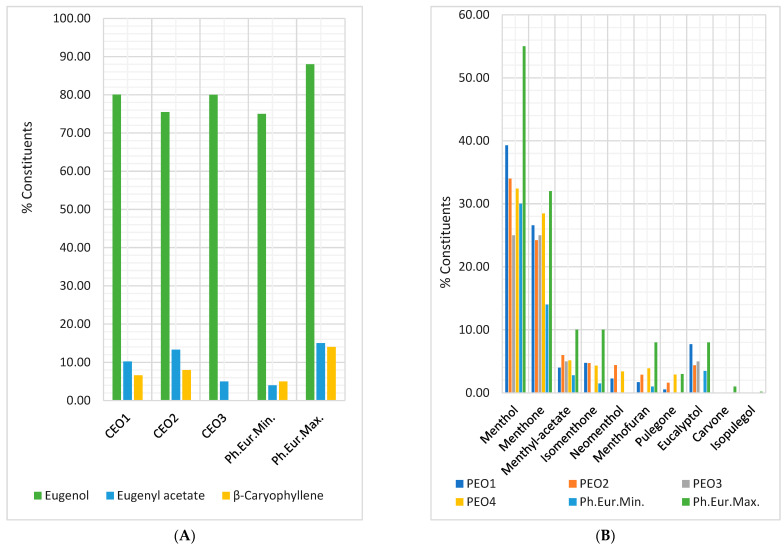
(**A**,**B**) Chemical composition of EO samples provided by the manufacturers, compared to Ph. Eur. 10 limits (Min/Max). (**A**) Clove oils (CEO1, CEO2, and CEO3); (**B**) Peppermint oils (PEO1, PEO2, PEO3, and PEO4); Ph. Eur.Min./Max.—the bioactive compounds content limits according to Ph. Eur. 10. (**C**,**D**) The correlation between chemical constituents and antibacterial and antibiofilm activities of Clove oil (**C**) and Peppermint oil (**D**) samples. S.a.—*S. aureus*, E.c.—*E. coli,* P.a.—*P. aeruginosa*, *ABE*—Antibacterial efficacy, *ABfE*—Antibiofilm efficacy.

**Table 2 antibiotics-12-01191-t002:** Antibacterial and antibiofilm efficacy of essential oils and antibacterial drugs against *S. aureus*.

		Antibacterial Effect			Antibiofilm Effect		
		D1	D2	D3	D1	D2	D3
GEN	Mean	87.33 ^x^	87.07	86.70 ^x^	50.60 ^a, b, x^	38.33 ^a, x^	37.90 ^b, x^
	SD	0.85	1.90	1.20	0.70	0.85	1.20
STR	Mean	88.00 ^y^	86.73	85.00 ^y^	48.27 ^a, b, x^	43.67 ^a, x^	41.47 ^b, x^
	SD	2.10	1.25	1.00	1.05	1.05	0.55
AMC	Mean	96.83 ^a, x, y^	89.66 ^a^	17.73 ^a, x, y^	72.87 ^a, x^	68.53 ^a, x^	17.43 ^a, x^
	SD	1.15	2.45	0.75	1.65	1.45	0.55
OEO1	Mean	92.76 ^a^	77.23 ^a^	12.70 ^a, x^	96.76 ^a^	96.33 ^b, x^	39.26 ^a, b, x^
	SD	2.25	1.55	0.80	1.75	1.65	0.75
OEO2	Mean	90.40 ^a^	78.03 ^a^	39.53 ^a, x^	94.20 ^a^	89.26 ^a, x^	19.13 ^a, x^
	SD	1.60	1.25	1.25	1.30	1.75	0.45
EEO1	Mean	88.50	86.80	86.46	53.63 ^a, x^	51.10 ^a, x^	ND
	SD	1.50	1.80	1.15	1.15	0.80	
EEO2	Mean	87.76	87.63	86.26	47.27 ^a, x^	35.10 ^a, x^	ND
	SD	1.65	1.67	1.62	1.25	1.40	
REO1	Mean	88.16 ^a^	87.13 ^b, x^	81.80 ^a, b, x^	40.27 ^a, x^	33.87 ^a, x^	28.10 ^a, x^
	SD	2.06	1.65	1.80	0.75	1.05	0.40
REO2	Mean	88.13 ^a^	75.26 ^b, x^	70.73 ^a, b, x^	49.23 ^a, x^	46.67 ^a, x^	18.33 ^a, x^
	SD	1.85	1.25	1.25	0.75	1.15	0.35
CEO1	Mean	91.26 ^a, x^	45.40 ^a, x^	40.13 ^a, x^	95.87 ^a, x, y^	95.40 ^b, x^	56.87 ^a, b^
	SD	1.75	0.90	0.85	1.85	1.80	1.85
CEO2	Mean	84.80 ^a, x^	51.63 ^a, x^	2.70 ^a, x^	81.23 ^a, x^	77.13 ^a, x^	ND
	SD	1.60	0.85	0.10	1.66	1.80	
CEO3	Mean	78.80 ^a, x^	47.23 ^a, x^	ND	83.50 ^y^	81.77 ^x^	ND
	SD	1.20	0.75		1.50	1.75	
PEO1	Mean	87.50 ^a, x, y^	82.43 ^a^	67.10 ^a, x^	91.13 ^a, x, y^	88.17 ^b, x, y^	77.20 ^a, b, x, y^
	SD	1.70	1.55	1.40	1.80	1.76	1.40
PEO2	Mean	85.23 ^a, z, w^	81.70 ^a^	56.83 ^a, x^	89.23 ^a, z, w^	88.67 ^b, z, w^	81.87 ^a, b, x, y^
	SD	0.45	0.80	0.95	1.25	1.65	1.85
PEO3	Mean	79.73 ^x, z^	ND	ND	32.17 ^a, x, z^	23.77 ^a, x, z^	18.27 ^a, x^
	SD	2.75			1.30	1.25	0.75
PEO4	Mean	79.83 ^y, w^	ND	ND	38.73 ^a, y, w^	20.33 ^a, y, w^	18.17 ^a, y^
	SD	2.75			1.15	0.65	0.70

Very good efficacy: ≥90%, good efficacy: 75–89%, moderate efficacy: 50–74%, satisfactory: 25–49%, and unsatisfactory: 0–24%. ND—Not detected; GEN—Gentamicin; STR—Streptomycin; AMC—Amoxicillin-Clavulanic acid; OEOs 1–2—Oregano essential oils from two different manufacturers; EEOs 1–2—Eucalyptus essential oils from two different manufacturers; REOs 1–2—Rosemary essential oils from two different manufacturers; CEOs 1–3—Clove essential oils from three different manufacturers; PEOs 1–4—Peppermint essential oils from four different manufacturers. For EOs, D1 = 25 mg/mL, D2 = 2.5 mg/mL, and D3 = 0.25 mg/mL. For standard antibiotics, D1 = 50 µg/mL, D2 = 5 µg/mL, and D3 = 0.5 µg/mL. The values followed by superscripts are statistically significant (*p* < 0.05). The differences were established between standard antibiotics and, also, between the samples of the same EO’ for each effect (antibacterial and antibiofilm). When the comparison was performed between decimal dillutions (D1, D2 and D3), the statistically significant values are marked with ^a^ and ^b^. When the samples effects were compared at the same dilution, the statistically significant values are marked with ^x, y, z^ and ^w^.

**Table 3 antibiotics-12-01191-t003:** Antibacterial and antibiofilm efficacy of essential oils and antibacterial drugs against *E. coli*.

		Antibacterial Efficacy			Antibiofilm Efficacy		
		D1	D2	D3	D1	D2	D3
GEN	Mean	89.26 ^x^	88.96	88.00 ^x^	76.33 ^a^	71.27 ^a^	63.90 ^a, x^
	SD	1.25	2.00	1.50	1.65	1.80	1.90
STR	Mean	89.13 ^y^	88.17	85.47 ^y^	72.90	67.87	67.77 ^y^
	SD	1.75	1.50	2.50	1.90	2.90	2.25
AMC	Mean	96.90 ^a, x, y^	84.47 ^a^	19.60 ^a, x, y^	82.47 ^a^	79.23 ^b^	52.47 ^a, b, x, y^
	SD	2.90	2.50	0.60	2.50	2.25	1.30
OEO1	Mean	94.00 ^a^	79.40 ^a, x^	13.40 ^a, x^	95.13 ^a^	86.10 ^a^	2.93 ^a^
	SD	2.00	1.80	1.40	3.15	2.45	0.45
OEO2	Mean	89.80 ^a^	68.90 ^a, x^	60.67 ^a, x^	95.63 ^a^	86.00 ^a^	ND
	SD	2.80	2.00	1.20	3.15	3.00	
EEO1	Mean	86.93	84.23	83.73 ^x^	61.73 ^a, x^	54.00 ^a, x^	ND
	SD	2.35	1.25	1.75	1.85	2.00	
EEO2	Mean	85.70 ^a^	85.47 ^b^	71.30 ^a, b, x^	52.33 ^a, x^	43.10 ^a, x^	ND
	SD	2.30	2.50	2.30	0.85	1.10	
REO1	Mean	85.77 ^a^	83.87 ^b^	71.57 ^a, b^	49.23 ^a, x^	42.50 ^a, x^	ND
	SD	2.75	3.85	2.10	1.25	0.90	
REO2	Mean	84.97 ^a^	82.30 ^b^	68.47 ^a, b^	30.07 ^a, x^	15.73 ^a, x^	ND
	SD	3.00	2.30	2.50	1.30	0.95	
CEO1	Mean	92.47 ^a, x, y^	46.27 ^a, x^	22.30 ^a^	95.10 ^a, x, y^	91.20 ^b^	11.20 ^a, b^
	SD	3.50	2.25	1.30	3.90	4.20	0.60
CEO2	Mean	78.10 ^a, x^	34.70 ^a, x^	ND	82.20 ^x^	ND	ND
	SD	2.10	1.70		3.20		
CEO3	Mean	76.80 ^a, y^	18.50 ^a, x^	ND	78.50 ^y^	ND	ND
	SD	2.80	0.50		4.50		
PEO1	Mean	79.00 ^a, x, y, z^	70.00 ^a, x^	45.57 ^a^	34.23 ^x, y^	ND	ND
	SD	3.00	2.00	2.50	1.55		
PEO2	Mean	72.50 ^a, x^	43.87 ^a, x^	29.80 ^a, x^	2.90 ^x^	ND	ND
	SD	2.50	2.15	1.80	0.10		
PEO3	Mean	72.77 ^y^	ND	ND	3.10 ^y^	ND	ND
	SD	2.80			0.10		
PEO4	Mean	71.30 ^a, z^	4.60 ^a, x^	ND	4.50 ^x, y^	ND	ND
	SD	2.30	0.20		0.15		

Very good efficacy: ≥90%, good efficacy: 75–89%, moderate efficacy: 50–74%, satisfactory: 25–49%, and unsatisfactory: 0–24%. ND—Not detected; GEN—Gentamicin; STR—Streptomycin; AMC—Amoxicillin-Clavulanic acid; OEOs 1–2—Oregano essential oils from two different manufacturers; EEOs 1–2—Eucalyptus essential oils from two different manufacturers; REOs 1–2—Rosemary essential oils from two different manufacturers; CEOs 1–3—Clove essential oils from three different manufacturers; PEOs 1–4—Peppermint essential oils from four different manufacturers. For EOs, D1 = 25 mg/mL, D2 = 2.5 mg/mL, and D3 = 0.25 mg/mL. For standard antibiotics, D1 = 50 µg/mL, D2 = 5 µg/mL, and D3 = 0.5 µg/mL. The values followed by superscripts are statistically significant (*p* < 0.05). The differences were established between standard antibiotics and, also, between the samples of the same EO’ for each effect (antibacterial and antibiofilm). When the comparison was performed between decimal dillutions (D1, D2 and D3), the statistically significant values are marked with ^a^ and ^b^. When the samples effects were compared at the same dilution, the statistically significant values are marked with ^x, y^ and ^z^.

**Table 4 antibiotics-12-01191-t004:** Antibacterial and antibiofilm efficacy of essential oils and antibacterial drugs against *P. aeruginosa*.

		Antibacterial Activity			Antibiofilm Activity		
		D1	D2	D3	D1	D2	D3
GEN	Mean	91.80 ^a^	58.97 ^a, x^	0.81 ^a, x^	86.40 ^a^	10.50 ^a, x^	ND
	SD	4.20	2.00	0.06	3.40	1.25	
STR	Mean	92.80 ^a^	88.93 ^b, x^	22.53 ^a, b, x^	81.90 ^a^	60.27 ^a, x^	ND
	SD	5.80	4.30	1.25	2.90	2.25	
AMC	Mean	91.40 ^a^	77.07 ^b, x^	29.50 ^b, x^	87.73 ^a^	76.73 ^a, x^	24.40 ^a^
	SD	4.40	2.90	1.50	3.75	2.75	1.40
OEO1	Mean	92.00 ^a^	70.73 ^a^	ND	96.00 ^a^	95.77 ^b^	27.77 ^a, b, x^
	SD	4.00	3.25		4.50	3.75	1.25
OEO2	Mean	91.40 ^a^	75.00 ^a^	28.00 ^a^	95.97 ^a^	90.83 ^b^	8.50 ^a, b, x^
	SD	5.40	3.50	1.50	4.05	5.15	0.60
EEO1	Mean	87.20	86.70	86.57	96.07	95.27	88.07
	SD	3.20	3.30	2.95	3.85	2.95	3.50
EEO2	Mean	87.90	86.47	84.93	96.10 ^a^	95.60 ^b^	86.30 ^a, b^
	SD	4.90	3.10	4.05	4.10	3.90	3.30
REO1	Mean	86.50 ^a^	84.70 ^b^	75.73 ^a, b^	94.93	94.07	93.87
	SD	3.70	3.80	3.75	2.95	2.90	2.23
REO2	Mean	87.50 ^a^	85.77 ^b^	66.10 ^a, b^	95.50	95.07	89.33
	SD	4.50	3.75	3.10	3.70	3.90	4.35
CEO1	Mean	93.60 ^a, x^	49.30 ^a, x^	21.90 ^a^	94.80 ^a, x, y^	92.40 ^b, x^	16.00 ^a, b, x^
	SD	4.58	2.92	2.02	4.47	3.04	0.76
CEO2	Mean	67.80 ^a, x^	20.50 ^a, x^	ND	43.90 ^a, x^	7.10 ^a, x^	4.10 ^a, x^
	SD	3.52	1.28		1.98	0.50	0.62
CEO3	Mean	59.60 ^a, x^	24.50 ^a, x^	ND	44.30 ^a, y^	16.40 ^a, x^	ND
	SD	2.53	1.54		2.38	1.59	
PEO1	Mean	86.70 ^a^	77.44 ^a^	69.70 ^a^	74.60 ^a, x^	68.80 ^b^	57.20 ^a, b, x^
	SD	4.15	3.87	2.97	3.51	2.89	2.59
PEO2	Mean	86.10 ^a^	79.20 ^b^	61.40 ^a, b^	73.20 ^a, y^	65.90 ^a^	48.40 ^a, x^
	SD	4.92	4.05	3.47	3.35	2.92	1.64
PEO3	Mean	86.10	ND	ND	79.90	ND	ND
	SD	4.51			3.94		
PEO4	Mean	85.00 ^a^	3.40 ^b^	ND	85.80 ^x, y^	ND	ND
	SD	4.44	0.18		4.05		

Very good efficacy: ≥90%, good efficacy: 75–89%, moderate efficacy: 50–74%, satisfactory: 25–49%, and unsatisfactory: 0–24%. ND—Not detected; GEN—Gentamicin; STR—Streptomycin, AMC—Amoxicillin-Clavulanic acid; OEOs 1–2—Oregano essential oils from two different manufacturers; EEOs 1–2—Eucalyptus oils from two different manufacturers; CEOs 1–3—Clove essential oils from three different manufacturers; PEOs 1–4—Peppermint essential oil from four different manufacturers. For EOs, D1 = 25 mg/mL, D2 = 2.5 mg/mL, and D3 = 0.25 mg/mL. For standard antibiotics, D1 = 50 µg/mL, D2 = 5 µg/mL, and D3 = 0.5 µg/mL. The values followed by superscripts are statistically significant (*p* < 0.05). The differences were established between standard antibiotics and, also, between the samples of the same EO’ for each effect (antibacterial and antibiofilm). When the comparison was performed between decimal dillutions (D1, D2 and D3), the statistically significant values are marked with ^a^ and ^b^. When the samples effects were compared at the same dilution, the statistically significant values are marked with ^x^ and ^y^.

## Data Availability

Data are available in the present manuscript and Appendix A.

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
