# Peer review of "Antibacterial and Antibiofilm Effects of Different Samples of Five Commercially Available Essential Oils"

_antibiotics, 2023, doi:10.3390/antibiotics12071191_

Round 1

Reviewer 1 Report

Dear authors,

greetings!

The manuscript “Antibacterial and Antibiofilm Effects of Different Samples of Five Commercially Available Essential Oils” uses principal component analysis to verify if samples of essential oils from oregano, eucalyptus, rosemary, clove, and peppermint presented antibacterial and antibiofilm activities and if those activities exhibited correlate with their chemical composition.

However, before publication, the manuscript needs minor adjustments.

Regarding “Introduction” in line 49 it is necessary to add “(EOs)" after “Essential oils”. In line 68 the information on market size needs a reference. Regarding “phytotherapeutic applications” (in line 115), it would be interesting to explore them, exemplifying the most important ones to each essential oil analyzed. It is crucial to also present in this section the motives for dedicating attention to the antibacterial activity of essential oils: Why was this specific activity chosen to be investigated?

Regarding results, Figures 1 to 4 and 6 lack standard deviation (SD) bars. As the experiments were performed in replicates, it is necessary to present SD to the readers. Figures 1,2, 3, and 6c also lack a y-axis title. Please, fix these aspects.

It would be interesting to analyze through CG-MS clove and peppermint essential oils to verify if the chemical composition is the one informed by the company. Adulteration can be responsible for the difference in activity.

Regarding the “Methods” section, there are some subtitles that should be in italics but that are not: 4.2.1 to 4.2.4. It is necessary to fix that.

Regarding the abbreviations used, it is necessary to standardize the manner to refer to the antibiofilm effect; ABfE (abstract, for example) or AbfE (line 139). When it comes to the Figures’ presentation, Figure 1 is duplicated in the manuscript appearing on pages 4 and 5.

Reviewer 2 Report

In this manuscript, the authors aimed to investigate the antibacterial and antibiofilm activities of five commercially available essential oils (Oregano, Eucalyptus, Rosemary, Clove, and Peppermint oils). Tedious analysis and detailed information were appreciated. The authors are suggested to highlight their scientific contribution, i.e., what is the new knowledge provided in this manuscript according to the detailed information.

Major

The antibacterial and antibiofilm effects of the essential oils examined in this study were reported previously as mentioned by the authors (Lines 354-355 with citations of [40-47]). Thus, instead of documenting antibacterial and antibiofilm effects of the essential oils as well as comparing the difference of some constituents among them, the authors are suggested to provide a link between the difference of antibacterial and antibiofilm activities of the five essential oils (Figures 1-3) and the difference of their constituents (Figure 6). Please point out which constituent(s) are the major active components responsible for the antibacterial and antibiofilm effects.

Minor: some writing revision such as,

  1. Lines 49: “Essential oils” should be “Essential oils (Eos)”  
  2. Figure 1: Figure 1A and 1B were redundant provided.
  3. Line 360: “[49][50]” should be “[49,50]”
  4. Many sections were composed of fragmented paragraphs (with only one sentence in a paragraph, in a way like itemized items), such as 3.6 General Considerations.

Reviewer 3 Report

Manuscript 2481626

Journal Antibiotics

Title Antibacterial and Antibiofilm Effects of Different Samples of Five Commercially Available Essential Oils

The manuscript entitled “Antibacterial and Antibiofilm Effects of Different Samples of Five Commercially Available Essential Oils” describes the antibacterial and antibiofilm effect of different batches of five essential oils (oregano, eucalyptus, rosemary, clove, and peppermint oils) against Gram-positive and Gram-negative bacteria. In addition, a correlation analysis through the PCA is reported. The manuscript is interesting but the structure and the presentation of the results needs substantial revision. Please see the comments in the file.

Some sentences should be revised

Round 2

Reviewer 2 Report

The authors have tried hard to revise the manuscript and adequately answered the questions I raised. Thus, the revised manuscript is recommended for publication.

Author Response

Dear Reviewer 2,

Thank you so much for your warm appreciation of our work.

We are grateful for your time, professionalism, and efforts to revise our manuscript, and we wish you all the best. 

Reviewer 3 Report

Authors addressed part of the reviewer's comments. Please follow minor comments below:

L19-42 Please rewrite the abstract. Please use a well-defined structure. The main quantitative data should be reported and the comparison among the sensitivity of different strains should be better included. Moreover, main conclusions of the work should be included at the end of the abstract.

L581-582 Please check this sentence. It is not correct. Revise

L591-592 Delete. It is not important.

L596-601 Please expand the discussion of these results with relevant

references.

L626-632 Please expand the discussion of these results with relevant

references.

L636-639 Please better discuss the antibiofilm action of REO and its link with the chemical composition, using relevant references.

L664-665 Delete. It is not important

L673-674 Please expand the discussion of this part

L716-719 Please better discuss these results with relevant references

L720-745 MIC values found in this manuscript are lower than MIC values reported in literature. Is this result due to the chemical composition of EOs or the microbiological assay used? Please explain in the text

L781-806 Please merge in one section.

Some sentences should be improved.
